# Coping with COVID-19: Differences in hope, resilience, and mental well-being across U.S. racial groups

Carol Graham[1,2]*, Yung Chun[3], Bartram Hamilton[4], Stephen Roll[3,5], Wilbur Ross[6], Michal Grinstein-Weiss[3,5]

**1** Global Economy and Development, The Brookings Institution, Washington, DC, United States of America, **2** University of Maryland, College Park, Maryland, United States of America, **3** Social Policy Institute, Washington University in St. Louis, St. Louis, Missouri, United States of America, **4** Olin Business School, Washington University in St. Louis, St. Louis, Missouri, United States of America, **5** Brown School of Social Work, Washington University in St. Louis, St. Louis, Missouri, United States of America, **6** Washington University School of Medicine in St. Louis, St. Louis, Missouri, United States of America

* CGRAHAM@brookings.edu

**Data Availability Statement:** These data were collected via a Qualtrics panel, and funded by the School of Social Policy at Wash University in St. Louis. It is available at a GitHub access link,

## Abstract

### Objectives

To explore if the COVID-19 pandemic revealed differences across racial groups in coping, resilience, and optimism, all of which have implications for health and mental well-being.

### Methods

We collect data obtained from four rounds of a national sample of 5,000 US survey respondents in each round from April 2020 to February 2021. Using logistic regression and fixed effects models, we estimate the pandemic impacts on COVID-19 related concerns, social distancing behaviors, and mental health/life satisfaction and optimism for racial/income groups.

### Results

Despite extreme income and health disparities before and during the COVID-19 outbreak, Blacks and Hispanics remain more resilient and optimistic than their White counterparts. Moreover, the greatest difference in resilience, optimism and better mental health—is found between poor Blacks and poor Whites, a difference that persists through all four rounds.

### Conclusions

These deep differences in resilience have implications for the long-term mental health of different population groups in the face of an unprecedented pandemic. Better understanding these dynamics may provide lessons on how to preserve mental health in the face of public health and other large-scale crises.

which provides the raw data files and variables that we used for our analysis in this paper, as well as the codes (https://github.com/SocialPolicyInstitute/SEICS/tree/main/PLoS-ONE). Access questions can be directed to Sarah Cowart at Wash U (scowart@wusl.edu).

**Funding:** The collection of the data used in this study was funded by Mastercard Center for Inclusive Growth, the JPMorgan Chase Foundation (63241517), the Annie E. Casey Foundation (220.5378), and Centene Corporation in the form of grants to MGW, SR, and YC. The funders had no role in study design and analysis, decision to publish, or preparation of the manuscript. No additional external funding was received for this study.

**Competing interests:** The authors have declared that no competing interests exist.

## Introduction

The outbreak of the Coronavirus disease 2019 (COVID-19) in the U.S. exposed deep vulnerabilities in our fragmented health care system as well as the broader consequences of extreme income inequality [1]. African American and Hispanic populations, who had the greatest income and wealth inequality compared with White populations before COVID-19 also suffered disproportionately high incidence and mortality rates from COVID-19 [2–4]. Inequalities in COVID-19 incidence and case mortality rates are well documented: while Black individuals make up just 12.5% of the U.S. population, they have accounted for 23% of COVID-19 deaths nationwide [5, 6]. Similarly, case rates and mortality rates have been higher for Hispanic and Latinx individuals across the country [7, 8].

One area about which less is known, however, is the degree to which the realities of COVID-19 have impacted people's physical and mental health, concerns and fears, and behaviors in response to COVID-19. There is reason to be concerned that COVID-19 may disproportionately impact these outcomes for racial and ethnic minorities. The disparity in the impact of COVID-19 across racial/ethnic groups occurred for several reasons, principally the longstanding untoward effects of institutional and structural racism. One notable manifestation of systemic racism is the overrepresentation of Black and Hispanic individuals in jobs deemed "essential" (e.g., in health, transportation, and service sectors), where working from home or maintaining social distancing is impossible [9].

Black and Hispanic people in the U.S. also have a higher probability of being low income and having worse access to good health care [10]. Long before the COVID-19 outbreak, these problems were exacerbated by systemic inequities in housing, health, employment, and opportunity [11, 12]. Racial and ethnic minority groups are also more likely to have comorbid conditions such as asthma, heart disease, and diabetes, all of which are risk factors for worse COVID-19 outcomes [10, 13–17].

Given such disparities, one might assume that minority and low-income populations would display greater fear of COVID-19 –potentially reflected in their social distancing and other behaviors, and the greatest losses on measures of health and mental well-being [18]. In this paper, we examine the intersection of social and economic factors that influence people's COVID-19 health behaviors. We investigate the relationship between how these individuals are coping with the pandemic and their mental well-being, as well as the ways in which the fear of COVID-19 influences health behaviors in non-Hispanic White, non-Hispanic Black, and Hispanic/Latinx individuals (hereafter White, Black, and Hispanic). Our findings are based on four waves of the Socio-Economic Impacts of COVID-19 Survey, fielded by a team of scholars, primarily from Washington University in St. Louis (and including these authors) and seeking to examine these issues in a diverse, national sample of U.S. adults. They reveal some surprises which include better reported mental health and well-being among racial/ethnic minorities throughout the pandemic, and complex patterns in the relationship between race/ethnicity and behavioral responses to it.

## Methods

### Data and sample

Data for are from the Socio-Economic Impacts of COVID-19 Survey. It is based on nationally representative survey samples from four waves of the survey, administered in late April—early May (Wave 1), late August (Wave 2), and November (Wave 3) in 2020, and March through April in 2021 (Wave 4) to approximately 5,000 nationally representative respondents in each wave, approximately half of which have repeated observations, forming a sub-set of

**Survey periods and daily COVID-19 cases**

**Fig 1. Source: New York Times (COVID-19 case), Oxford University (Vaccinations).**

respondents that are a panel (Fig 1). The survey relied on Qualtrics online panels developed using quota sampling techniques to ensure that the sample approximated United States demographic characteristics in terms of age, gender, race/ethnicity, and income. (Online, non-probability samples using Qualtrics panels typically generate samples that closely approximate those of the General Social Survey, which is considered the gold standard in survey administration [19]).

The survey was submitted to the Washington University at St. Louis IRB (ID: 202004100) and was approved. This was not a requirement as it is an on-line internet survey with voluntary participation, but we sought the additional approval. We received written consent. Even though it was exempt, we still got written (online) consent from all research participants before they entered the survey. The survey was also limited to adults. A copy of the most relevant questions from the 4 waves of the survey, utilized here, are in included in the supplementary materials. The data, variables, and codes can be found at this link: (https://github.com/SocialPolicyInstitute/SEICS/tree/main/PLoS-ONE).

The overall response rate across four waves of the survey was 10.1%, and ranged from a low of 6.8% in Wave 4 to a high of 13.5% in Wave 3. (Response rates were calculated using the American Association of Public Opinion Research's RR2 measure [20]). After exclusions due to quota restrictions and quality checks embedded in the survey, 22,444 respondents remained in the sample. (For the comparison between the survey sample and the ACS 2019 estimate [21], see Table 1). Additional checks on the characteristics of this sample revealed that it also approximated the U.S. population in terms of state of residence, in addition to the quotas specified above. Finally, we excluded respondents who did not provide a response to the items used in this analysis or who did not identify as White, Black, or Hispanic—the racial/ethnic populations of interest in this study—resulting in a final analytical sample of 16,680.

**Table 1. Survey sample and ACS 2019 sample comparison.**

| Characteristic | 2019 ACS* (1-Year Estimates) | Study sample (Wave 1 only) | Study sample (Wave 1 to 4) |
|---|---|---|---|
| Age | 50.4 | 46.6 | 43.1 |
| Race/Ethnicity | | | |
| White, Non-Hispanic (%) | 60.7 | 61.7 | 62.0 |
| Black, Non-Hispanic (%) | 12.3 | 12.2 | 12.5 |
| Asian, Non-Hispanic (%) | 5.5 | 5.3 | 5.5 |
| Hispanic (%) | 18.0 | 17.4 | 17.4 |
| Other (%) | 4.3 | 2.4 | 2.7 |
| Male (%) | 48.5 | 49.6 | 47.2 |
| Gross Annual Household Income | | | |
| Less than $25,000 (%) | 18.0 | 16.3 | 21.0 |
| $25,001 - $50,000 (%) | 20.4 | 22.7 | 21.7 |
| $50,001 to $75,000 (%) | 17.4 | 16.5 | 18.6 |
| $75,001 to $100,000 (%) | 12.8 | 14.4 | 13.1 |
| More than $100,000 (%) | 31.4 | 30.2 | 25.6 |

**Notes:** * We limit the sample to 18 or older.

## Measures

To measure life-satisfaction, optimism, and mental health, we utilized two sets of questions. The first was a standard life satisfaction question (the Cantril ladder), which asks: "Please imagine a ladder, with steps numbered from 0 at the bottom to 10 at the top. The top of the ladder represents the best possible life for you, and the bottom of the ladder represents the worst possible life for you. On which step of the ladder would you say you personally feel you stand *at this time* [22]?" A subsequent question asks the respondent where they think they will be on the same ladder in *five years*. The second question is designed to capture optimism/hope for the future in both the near-term and the long-term (for detail, see Graham et al. [23]). The second measure of well-being asks respondents two questions assessing their mental health on a five-point response scale ranging from poor to excellent. The first question asks respondents to assess their mental health *three months ago*. The second asks them to rank their mental health *currently* (i.e. at the time of each wave).

To measure COVID-19 related fears among survey respondents, the survey asked "how afraid are you of the COVID-19 pandemic?" where respondents could indicate their level of fear on a scale of 0 (not afraid) to 100 (very afraid). To explore social distancing behaviors during the pandemic we asked respondents to report the degree to which three statements about social distancing practices described their own behaviors. These statements concerned wearing a mask, avoiding social gatherings, and notifying the people around them if they exhibited COVID-19 symptoms. Each response ranged from 0 ("does not describe me") to 100 ("describes me very well"). These measures were adapted from the "Measuring Worldwide COVID-19 Attitudes and Beliefs" projects [24].

We examined each of the above outcomes by respondents' race/ethnicity and income level. For the purposes of this study, we only present estimates for White, Black, and Hispanic respondents, as it is between these groups that the largest reported well-being disparities have been observed in other research [25, 26]. We constructed the income groups in this study as a function of self-reported annual household income in 2019, household size, and the U.S. Department of Housing and Urban Development' measure of area median income (AMI) at the county level. Based on HUD's methodology for determining the eligibility of applicants for

assisted housing programs [27], we defined three income cohorts based upon the AMI at the country level, adjusting for family size: low income: [0, 80% AMI); moderate income: [80, 120% AMI); and middle and high income: [120% AMI,.).

In addition to the two key explanatory variables of race/ethnicity and income, we controlled for a set of covariates including demographic characteristics, health insurance status, the self-reported experience of COVID-19 symptoms, the prevalence of COVID-19 cases in the county in which the respondent lived as of the date of survey response, the population density of the respondent's county of residence, and the respondent's Census division. Demographic characteristics included gender, age, marital status, the number of children in the household, and educational attainment. In later waves, we also added in questions about likelihood of vaccination, financial shocks due to COVID-19, and other questions related to change in employment and health status due to the virus.

## Statistical analysis

We estimated the relationships between race/ethnicity, income, and the array of outcome measures specified above using Ordinary Least Squares (OLS) regression models, with heteroscedasticity-robust standard errors. In addition to controlling for all the variables outlined above, we also included interaction terms for race/ethnicity and income, which allowed us to estimate the joint relationships between these variables and the outcomes of interest. For the sake of simplicity, we primarily focused on disparities between the lower- and middle/high-income cohorts. The data analysis in this study was conducted using Stata (version 16) and we used a statistical significance threshold of $p < .10$.

## Results

### Sample description

Overall, the sample well represents the U.S. population with respect to gender, age and marital status, and racial/ethnic composition. However, compared to the U.S. population, our sample is more highly educated; 57% of our respondents held Bachelor's degree or higher, which is much higher than the U.S. population (32%) [28]. The proportion of respondents without dependents under 18 years old is 74%, which is slightly higher than the U.S. population (69%) [28]. About three in five of our respondents were considered to be low- and moderate-income, whose annual income was less than 120% AMI in 2019 at the country level. Table 2 reports the demographic characteristics of our sample.

### Life satisfaction, optimism, and mental health

We explore our respondents' life satisfaction, mental health, and optimism during the pandemic. Figs 2 and 3 plot the changes in respondents' current life satisfaction and their expected life satisfaction 5 years in the future—a proxy for optimism—by race/ethnicity and income. Fig 4 plots the change in respondents' mental health throughout the pandemic by race/ethnicity and income.

Higher incomes were associated with higher levels of life satisfaction, optimism, and mental health during the pandemic. In addition, we observe higher/better levels of these metrics among Black respondents, and smaller but still significant differences in life satisfaction and optimism among Hispanic respondents, with a slight temporary uptick in August of 2020 among rich Hispanics which we cannot fully explain. The differences in reported life satisfaction and optimism for the future between Black and White respondents were roughly as large as the differences between higher and lower income groups, and the gaps in optimism between the two groups were largest at low-income levels. Black respondents also reported better

**Table 2. Analytic sample characteristics.**

| | Percent |
|---|---|
| **Gender:** | |
| Non-Female | 49.1% |
| Female | 50.9% |
| **Age***: | |
| 18–24 | 11.0% |
| 25–34 | 19.3% |
| 35–44 | 16.4% |
| 45–54 | 17.8% |
| 55+ | 35.6% |
| **Marital Status:** | |
| Single ** | 39.1% |
| Married or living with a partner | 60.9% |
| **Number of Dependents:** | |
| No child | 72.1% |
| 1 Child | 13.5% |
| 2 Children | 10.6% |
| 3 Children or More | 3.8% |
| **Education** | |
| High School or Lower | 14.9% |
| Some College/Associate's Degree | 30.8% |
| Bachelor's degree | 29.6% |
| Graduate/professional degree | 24.7% |
| **Health insurance** | |
| Have a health insurance | 91.3% |
| Not have a health insurance | 8.7% |
| **Race/Ethnicity** | |
| White, non-Hispanic | 67.4% |
| Black, non-Hispanic | 14.2% |
| Hispanic | 18.4% |
| **Income** | |
| Low Income [0, 80% AMI) | 44.3% |
| Moderate [80%, 120% AMI) | 20.1% |
| Middle, and High Income [120% AMI,.) | 35.7% |
| **Total** | 16,680 |

Notes: Reference categories are underlined

* In the analytic model, we use age and age squared as continuous variables

** Single includes never married, separated, divorced, and widowed.

mental health than Whites. We did not observe any significant interactions between income level and race/ethnicity. What is most remarkable is that these trends remained remarkably stable throughout the pandemic. While there were modest drops in optimism and in mental health status at a few peaks, such as November 2020, the overall trends and the gaps across races remained similar, with Blacks in general and low income Blacks in particular retaining higher levels of optimism and lower levels of stress throughout the pandemic.

New questions about feeling anxious, nervous, and weary were added to the survey in the final three waves. The patterns in these responses are again consistent with lower levels of

## Life satisfaction

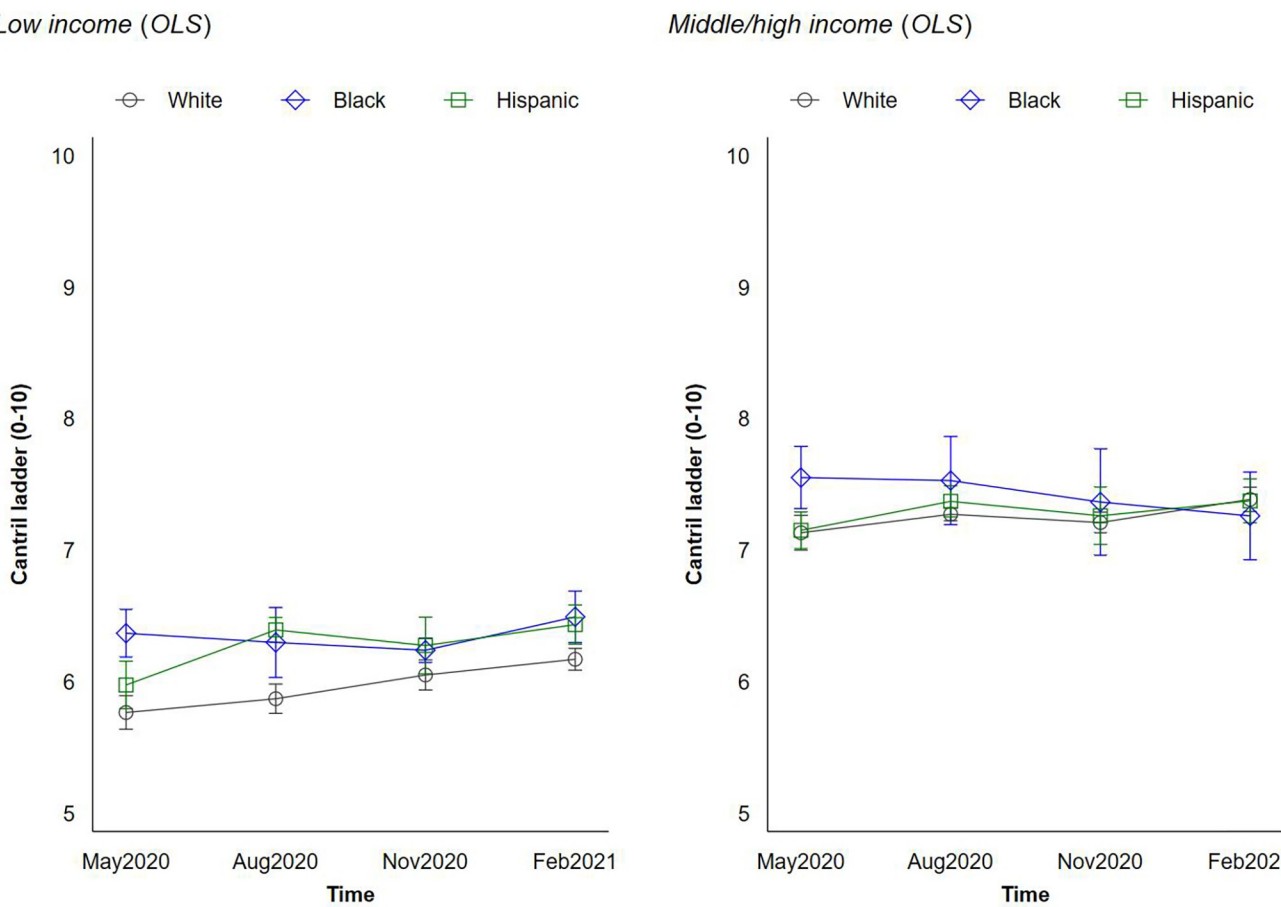

90% confidence interval

**Fig 2. Changes in life-satisfaction by race and income.**

anxiety for Blacks and in particular low-income blacks, than for other races and particularly compared to Whites. We ran an additional model, based on the PHQ-4 index (Fig 5), which sums up four indicators gauging depression (i.e., anxiety, worrying, weariness, and hopelessness), and is widely used in the clinical mental health field as a tracking tool (*not* a clinical diagnostic one). This is appropriate for our purposes, meanwhile, as we are following trends in reported depression and anxiety, not clinical diagnoses.

We first find that patterns in overall depression/anxiety are relatively constant over the course of the pandemic. Comparing across incomes, meanwhile, we find that higher income respondents exhibited lower anxiety and depression levels in general than the others through the pandemic, with levels in the "normal" range. Lower income respondents had mild levels of anxiety and depression, meanwhile. Across race/income groups, low-income Blacks again exhibit significantly lower levels of anxiety and depression than either low-incomes Whites or Hispanics, and these differences are large enough to drive the average overall trend in which Blacks have lower levels of anxiety and depression.

## Optimism, 5 years after

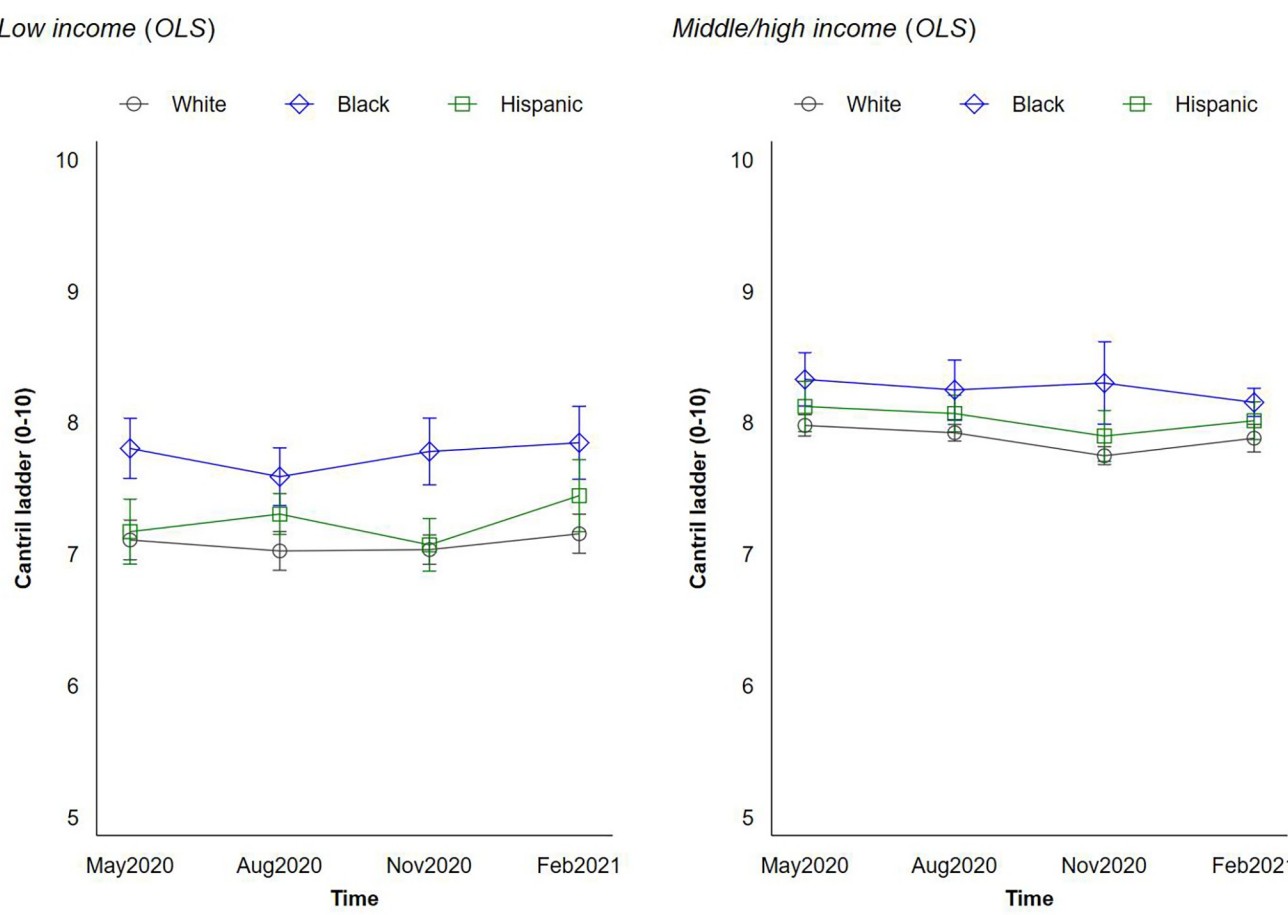

90% confidence interval

**Fig 3. Changes in optimism by race and income.**

### COVID-19 related fears/concerns

Overall, White respondents exhibited lower levels of COVID-19 fear than Black and Hispanic respondents and reported lower expected probabilities of COVID-19 infection and death than Hispanic respondents. This pattern remained quite consistent throughout the four survey waves. Fig 6 plots the predicted change in COVID-19 fear scores. Throughout the pandemic, the levels of COVID-19 related fears have not drastically changed except for White respondents—both lower- and higher income White respondents reported lower fear levels in comparison with those in the early period. Notably, the fear level of Black respondents with lower income was drastically increased during the summer in 2020, which might associated with the Black Lives Matter movements in the same period. With respect to variations by income, White respondents reported significant and negative associations between COVID-19 fear and income levels over the course of the pandemic. However, higher-income Black respondents reported significantly higher levels of COVID-19 fear than those with lower incomes, perhaps because of higher health literacy, better information, and more awareness than their lower income counterparts [29].

## Mental health

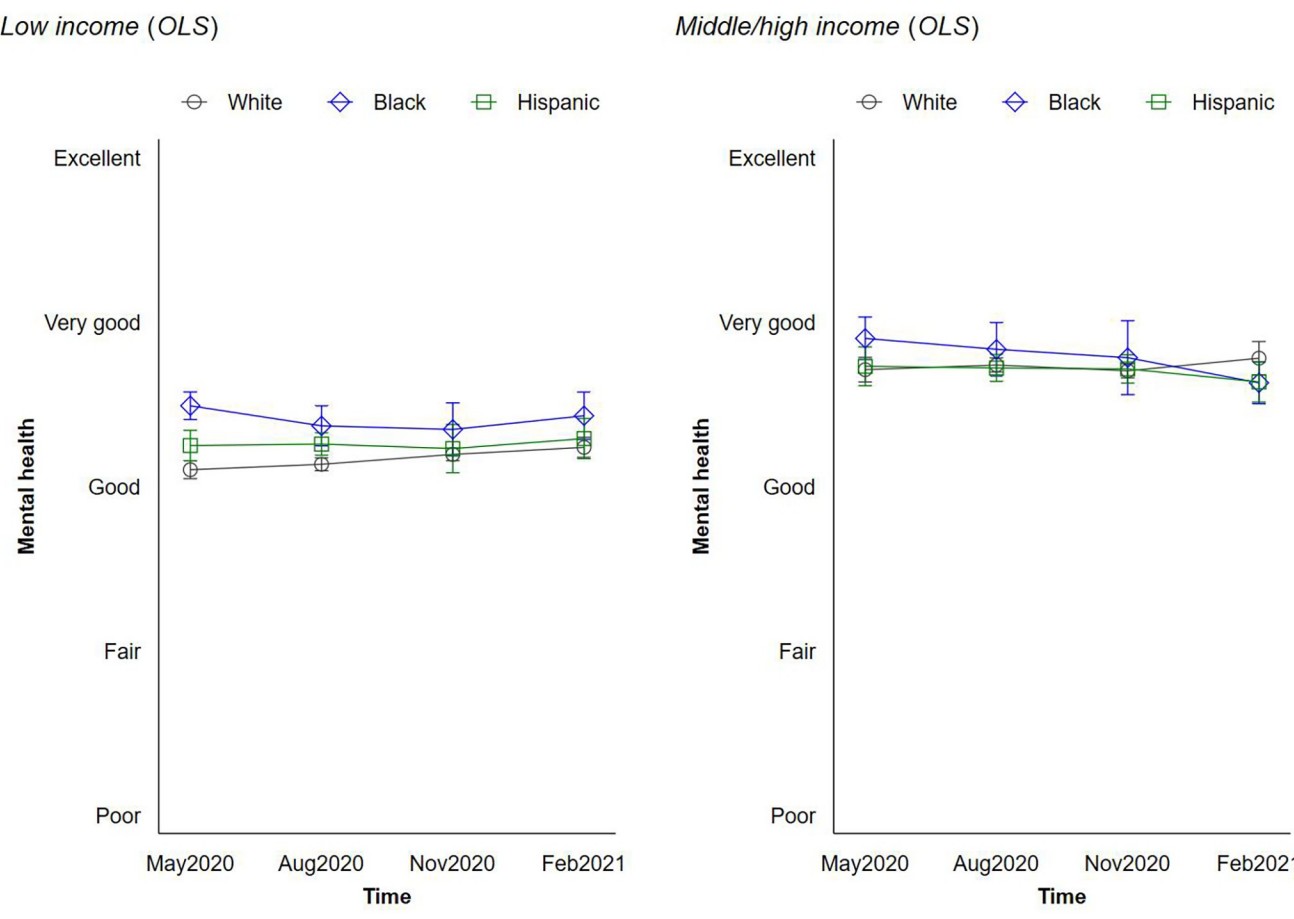

90% confidence interval

**Fig 4. Changes in mental health by race and income.**

It is possible that fear reports are influenced by traits such as optimism and pessimism and, as noted above, low income Blacks are more optimistic than most other groups. We explored the correlation between life satisfaction and COVID-19 fears. It is modestly negative, at 0.554, with a P-value of .003. This is intuitive but not large enough to drive the findings in a substantial way.

### Social distancing behaviors

Finally we explored the changes in social distancing behaviors by respondents' race/ethnicity and income. These were assessed by wearing a mask (Fig 7), avoiding social gatherings (Fig 8) and informing one's COVID-19 related symptoms (Fig 9). White respondents were less likely to wear masks than Hispanic respondents regardless of income level, though White and Black households were similarly likely to wear masks. Self-reported mask-wearing scores stay flat as income increases for White and Hispanic respondents. While the score for mask wearing for Black respondents increases as their income level rises, this difference is not statistically

## PHQ-4

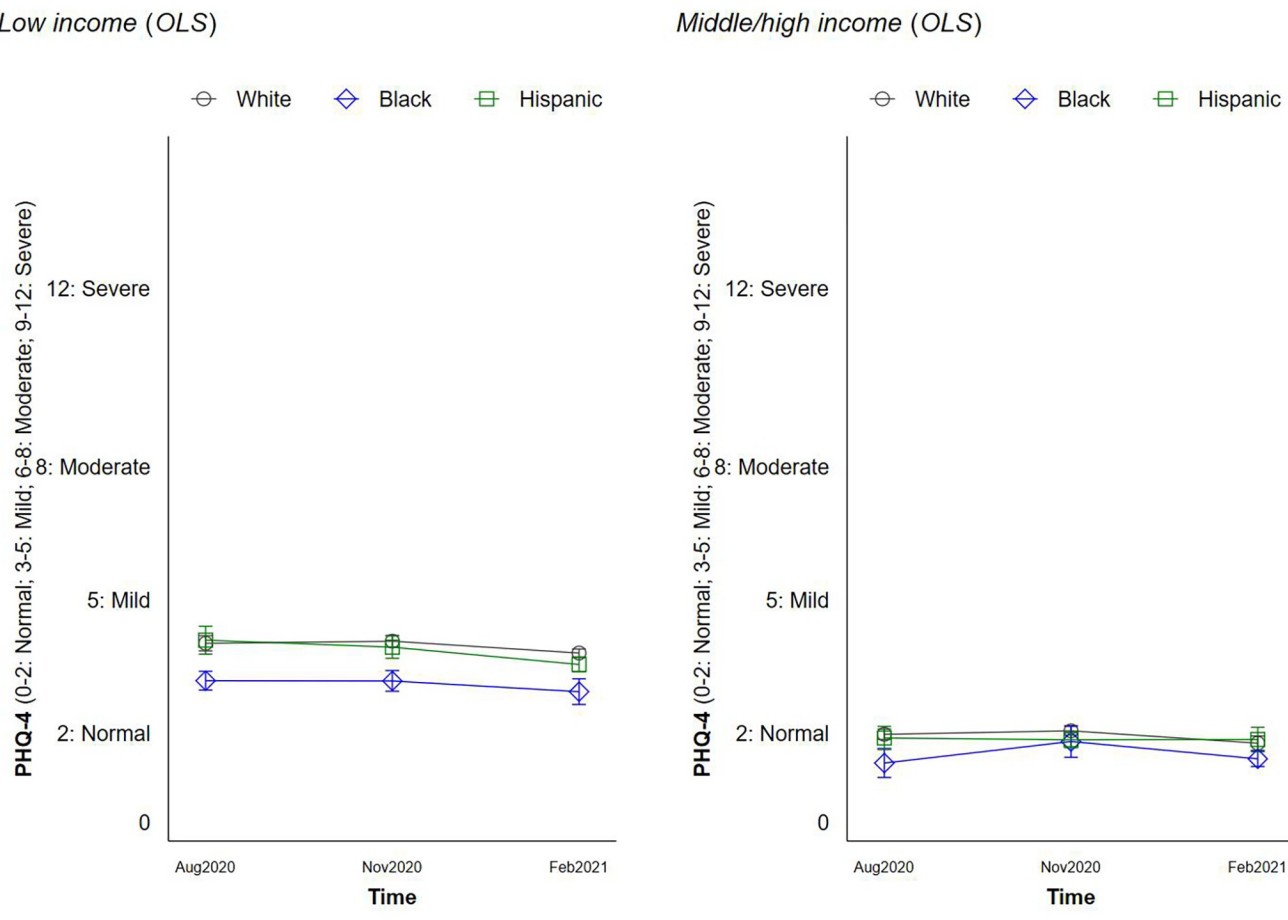

90% confidence interval

**Fig 5. Changes in PHQ4 by race and income.**

significant. Over the course of the four waves, meanwhile, the difference in mask wearing across races narrowed quite a bit.

Overall, Black respondents reported a significantly lower propensity to avoid social gatherings than White and Hispanic respondents. This difference appears to be driven by income, however, since higher-income Black households were as likely as those in the other two high income groups to report avoiding social gatherings. The lower reported rates of avoiding social gatherings for the low- and moderate-income (LMI) Black respondents might be associated with their employment status; 76% of Black LMI respondents reported that they were required to be physically present at their primary place of employment, which was significantly higher than the proportion of White (70%) and Hispanic (68%) LMI respondents. Their propensity to avoid gatherings increased slightly though in later waves of the survey, meanwhile. The narrowing of gaps across race/income groups likely reflects more acceptance of protective behaviors as the virus spread quite dramatically across the country (with, of course, some groups still resisting the restrictions).

## COVID-19 related fear

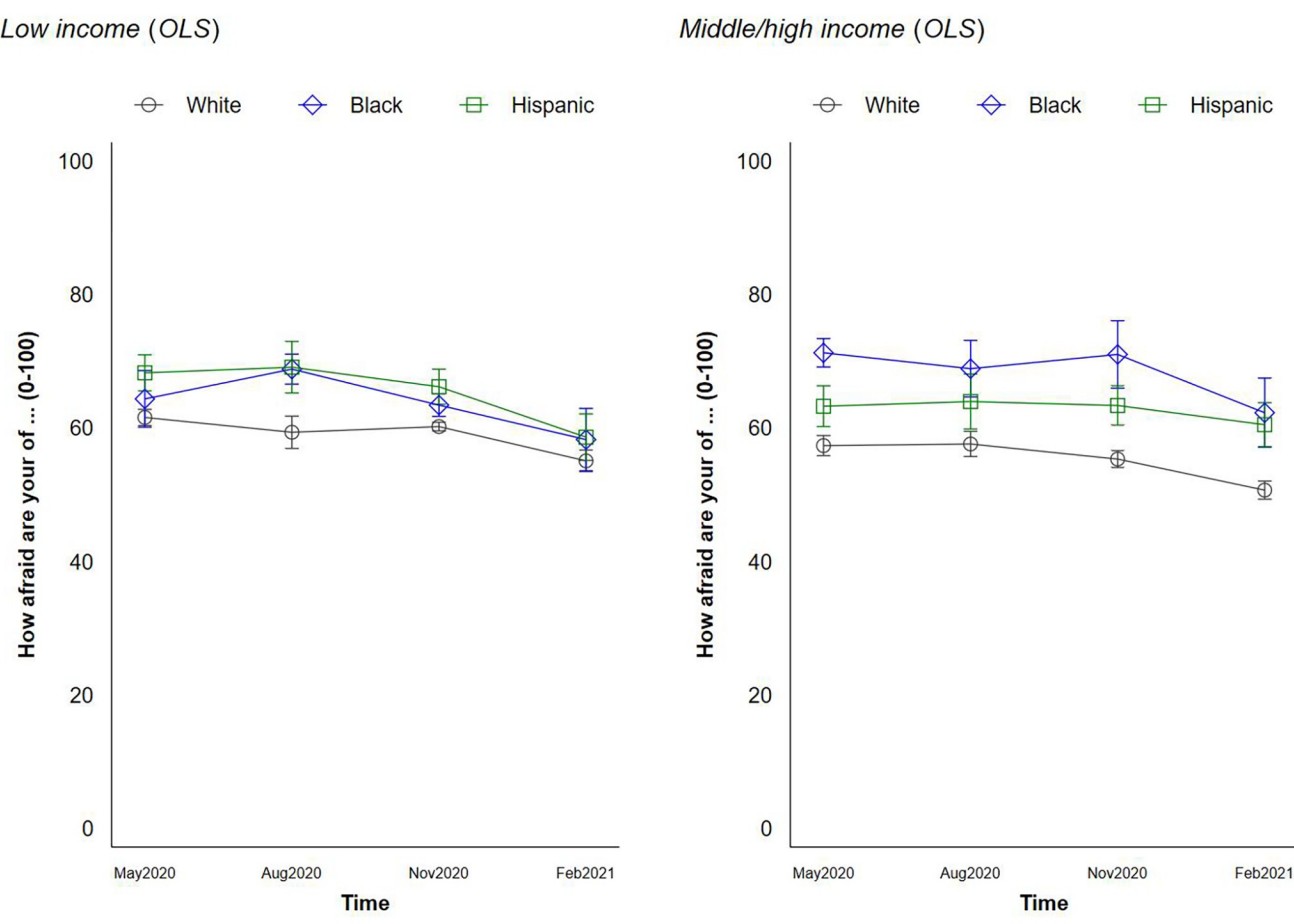

90% confidence interval

**Fig 6. Changes in COVID-19 related fears by race and income.**

Higher-income respondents were more likely than lower-income respondents to report that they would inform others of any COVID-19 symptoms they exhibited, and low income Black households were the least likely to report that they would inform others of their symptoms than those in other groups, perhaps because of fear of job loss and/or greater mistrust of a health system that has systematically discriminated against them.

## Discussion

The COVID-19 pandemic has exposed deep vulnerabilities across our fragmented health care system and across the income distribution. At the same time, it presents an urgent opportunity to better understand—and ultimately address—the factors driving health behaviors and persistent health inequities.

While it may be tempting to blame differences in COVID-19 infection rates on individuals' health behaviors, the answer to resolving COVID-19 disparities is more nuanced. Although personal compliance with guidelines such as handwashing, wearing a mask, and social

## Wearing a mask

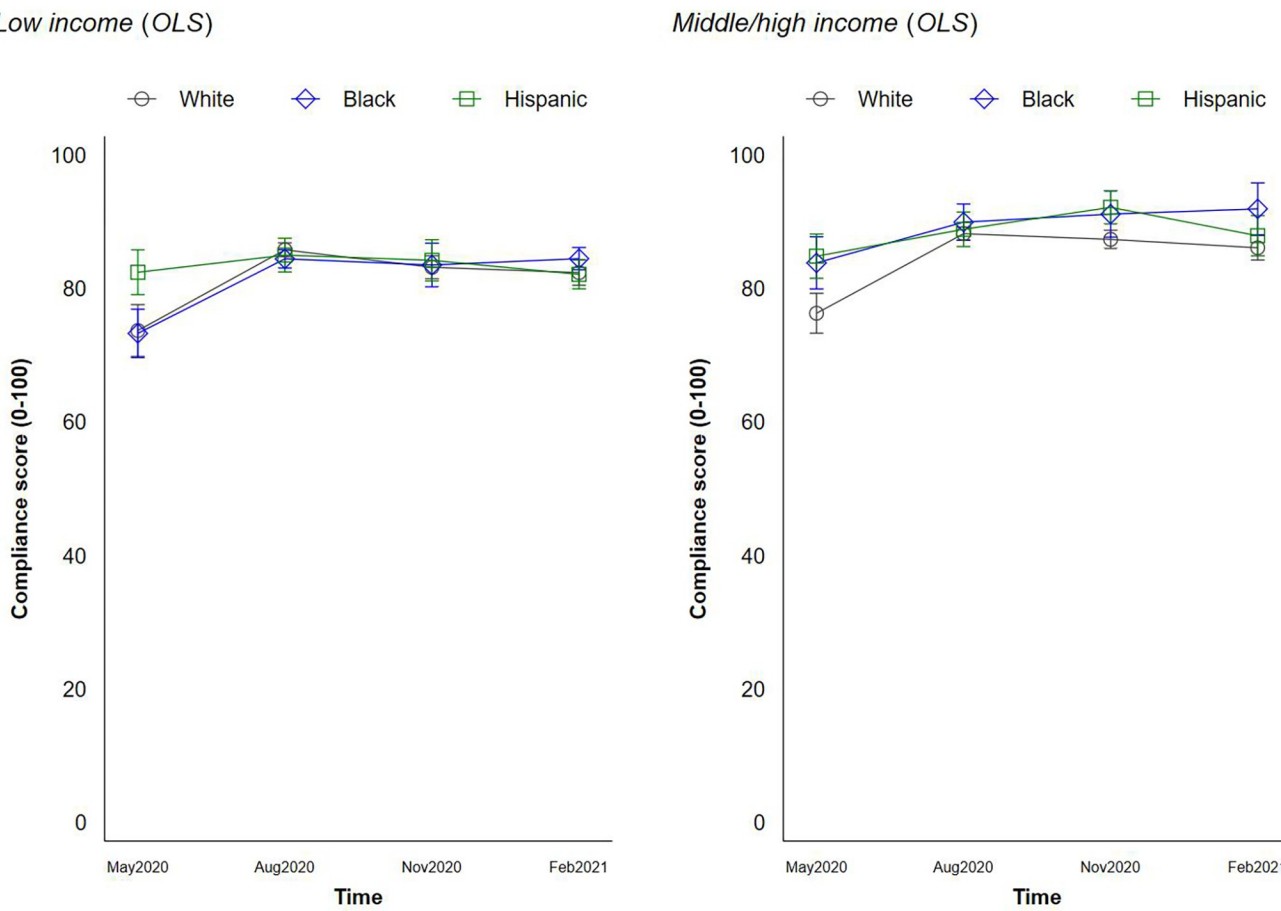

90% confidence interval

**Fig 7. Changes in social distancing behaviors by race and income–wearing a mask.**

distancing is critically important to reducing the spread of COVID-19, the pandemic has highlighted structural factors—who is designated an essential worker, the types of work they do, and whether they get sick days—and their profound influence on the health of individuals and the health of subgroups within the U.S. population. As such, the intersection of structural racism, social and economic factors influence people's COVID-19 health behaviors. We explored this intersection by looking specifically at the ways in which the fear of COVID-19 influenced health behaviors as well as the impact of coping with the pandemic on mental well-being.

### Role of fear in shaping health behaviors

Since the outbreak of the COVID-19 pandemic, the public has been bombarded by nearly continuous media updates of the threat of coronavirus, increasing infection rates, and new milestones in death counts. Living with this constant threat can increase anxiety and have

## Avoiding gatherings

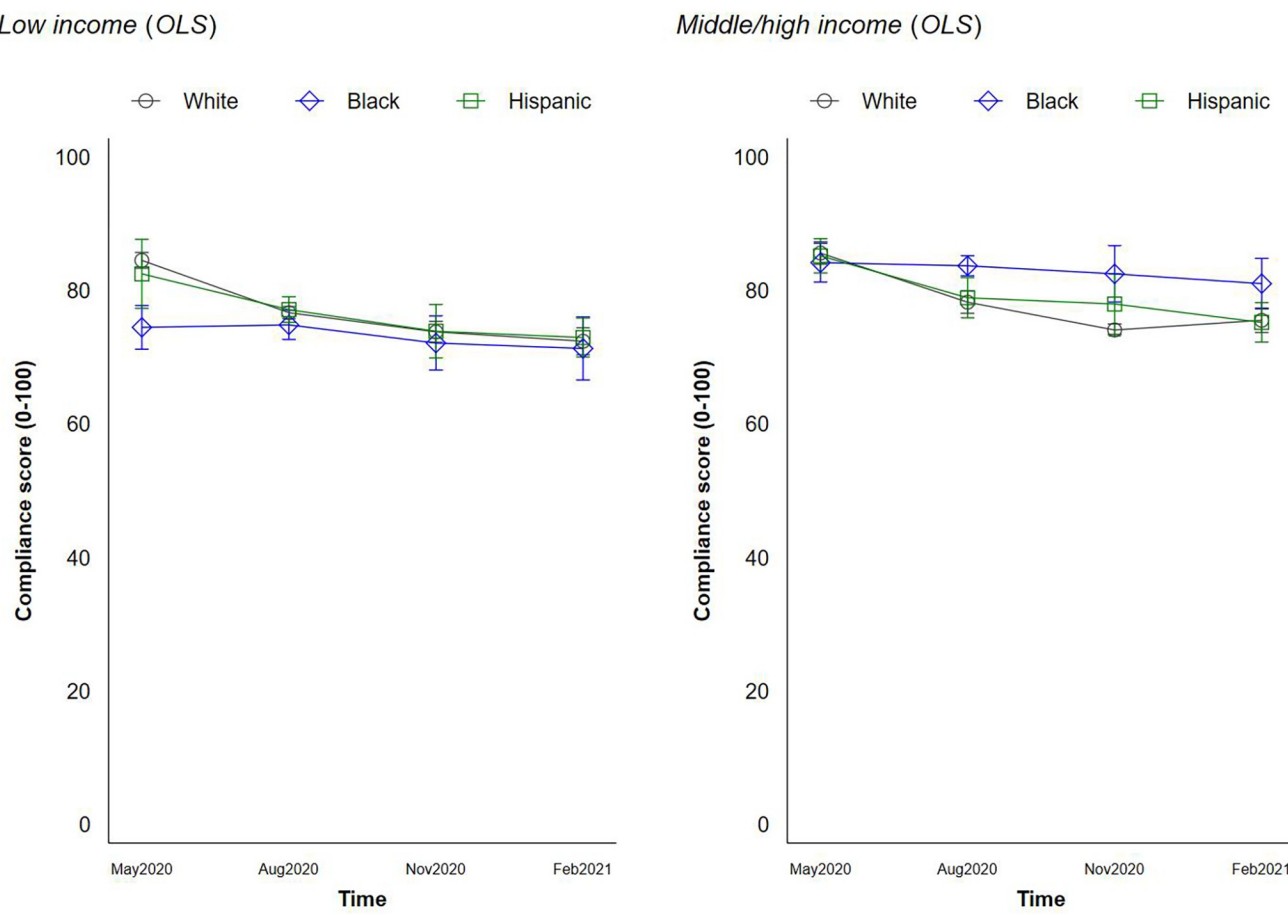

90% confidence interval

**Fig 8. Changes in social distancing behaviors by race and income–avoiding social gathering.**

immediate negative effects on mental health. Although the fear of infection could be expected to manifest itself in cautious, careful behavior aimed to prevent acquiring the illness, such fear can be overridden by other factors such as the messaging people receive about the disease, perception of personal infection risk, and economic factors such as income and whether a worker has paid sick days. With other widespread infections or epidemics such as the 2013–2016 West Africa Ebola virus outbreak, the public's fear of infection led to behaviors that followed expected patterns of being conformist and less accepting of individualistic behavior [30]. COVID-19 has certainly elicited such behaviors in the U.S. population. However, in contrast to fear-driven behavior observed in other epidemics, coping with COVID-19 has brought about unexpected behavioral patterns ranging from harsh social attitudes, more conservative attitudes toward immigration, and even swaying political opinion and affiliations in which strong individualistic attitudes jeopardized the effectiveness of medically proven guide-lines– such as mask wearing–for containing the spread of the disease.

## Informing COVID-19 Symptoms

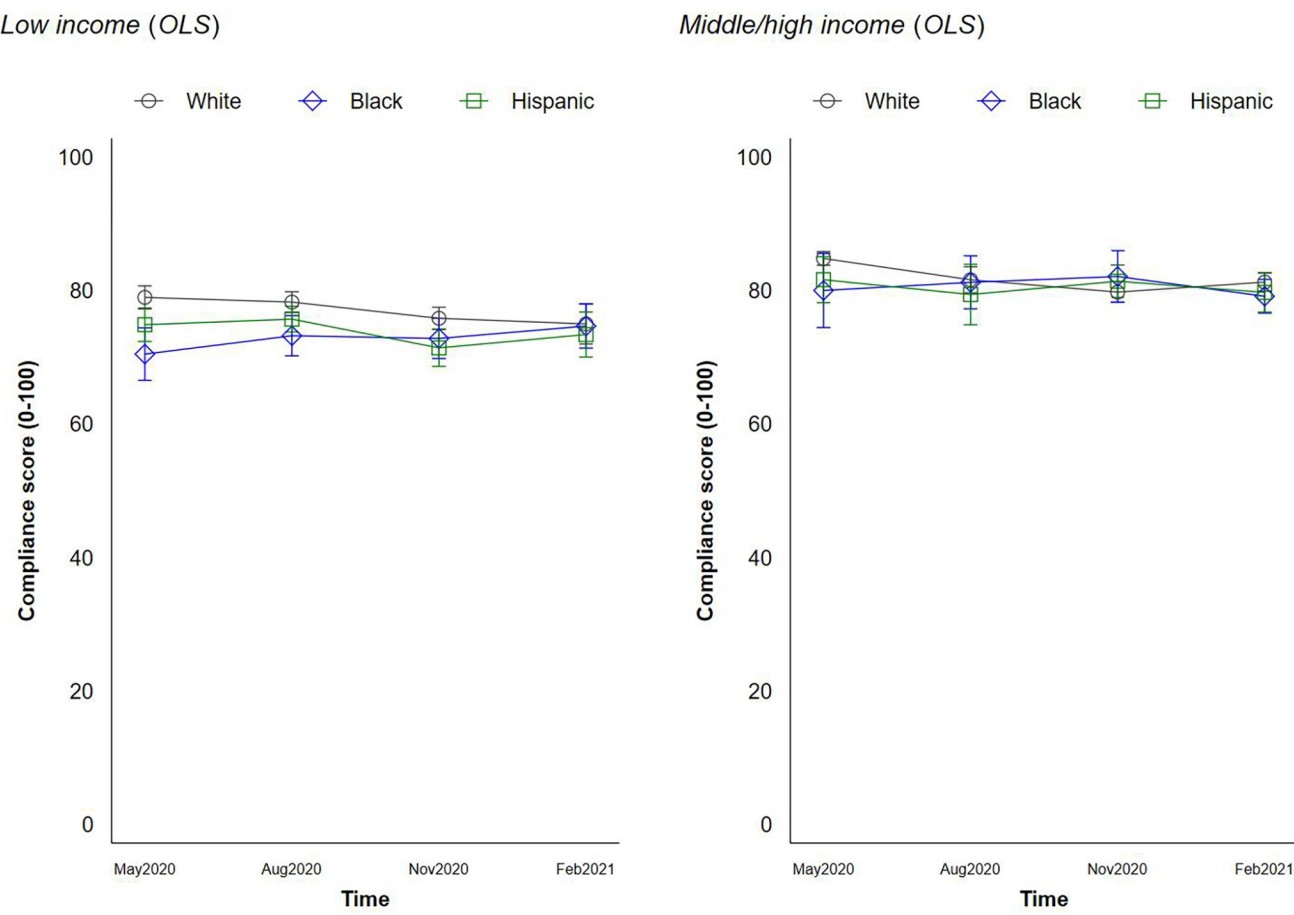

90% confidence interval

**Fig 9. Changes in social distancing behaviors by race and income–informing one's COVID-19 related symptoms.**

### Role of fear in COVID-19 related health behaviors

Given standard patterns of health behaviors and the social determinants of health in which higher levels of income are associated with better access to care and better health outcomes, we were not surprised to find similar results in our sample. Across the intersection of three racial groups and income levels, White respondents with high levels of income reported the lowest levels of fear of COVID-19. However, we did not expect to find that Black respondents with high income levels would be more likely than low-and middle-income Black respondents to report experiencing high levels of fear related to COVID-19.

What factors cause affluent White respondents to be insulated from COVID-19 fear whereas their affluent Black counterparts appear to experience high levels of COVID-19 fear? While racial and ethnic background does not protect anyone from COVID-19, the responses of affluent White respondents may reflect greater confidence in their ability to access medical care when needed as well as higher trust in the health care system [18]. The attitudes of more affluent Black respondents may reflect a greater sensitivity to the disproportionate rates of

COVID-19 deaths among the Black population in general. Further, for affluent Black respondents, the COVID-19 disparity likely underscored historic health disparities and the ways in which Black patients are often marginalized within the health care system, no matter their wealth levels. Wealthy Blacks may be more aware of these glass ceiling effects than poorer ones. Assari [31], for example, finds that while increasing levels of education have a protective effect on mental health on average, that effect is lower for Blacks than for other races. Additionally, COVID-19 fear among low-income Blacks may be masked by an avoidance strategy that accepts realities that would normally induce toxic, traumatic stress. Acceptance of one's fate, and in this case, denial of fear, has been documented to offer a long-term survival advantage [32]. However, in the context of COVID-19 related illness, acceptance does not connote an acceptance of racist constructs that consigned Blacks and other people of color to suffer higher morbidity and mortality from COVID-19. While adaptive in the short term, acceptance and avoidance is inherently counter-productive. Wilson and Murrell [33] describe the ways that both avoidance of our emotions and avoidance of meaningful contexts contribute to the maintenance of stress and anxiety.

Fear of infection is likely hardwired in the human race to drive behaviors that benefit the species. If so, the fear driver should result in behaviors that are cautious and aim to limit personal and group risk of infection. However, we did not find fear of COVID-19 was consistently associated with better compliance with health guidelines such as wearing a mask in public or limiting social contacts. Indeed, our findings seem puzzling at first because Black respondents, some of whom reported the highest levels of COVID-19 fear, initially reported a much low propensity to avoid social gatherings. Yet a closer look at the intersection of race and income reveals that higher-income Black respondents were equally likely as their White counterparts to avoid social gatherings, whereas lower income Blacks were significantly less likely to. The differences in avoiding social gatherings between low income Blacks and other racial groups also narrowed quite a bit across the survey waves and the virus began to subside in many places by early in 2021, and people of all races began to socialize more.

These differences are likely based in the economics of racial inequity, with Black individuals overrepresented in low-wage jobs that require them to be physically present in their workplace and less able to avoid social gatherings [34]. Although deemed essential workers, these workers are often paid minimum wage and do not receive benefits such as paid sick leave. Frequently, the combination of low pay and no benefits means they do not have the choice to practice social distancing, as few can afford to forgo a paycheck and stay at home. The preference for social gatherings among lower income Blacks also a cultural phenomenon, which emphasizes humanity as communal rather than individualistic [35, 36].

Further, whereas affluent respondents indicated a greater likelihood to self-report if they have symptoms consistent with COVID-19, lower-income Black respondents were significantly less likely to self-report their COVID-19 symptoms than other low-income groups. Again, this difference in health behavior is likely explained by the opportunities and choice afforded to affluent respondents. Those with higher incomes are more likely to work in jobs that allow some or all of their work to be done remotely [37]. In contrast, those at the other end of the economic spectrum are often living paycheck-to-paycheck and often report feeling they do not have a choice in continuing to work when feeling ill because their household is dependent on their paycheck. Low-income Blacks may also be less trusting than other cohorts of accessing good quality health care [38]. Again, though, the gaps narrowed significantly across races towards the final wave of the survey, with Hispanics less likely to report symptoms than Blacks (likely for similar reasons, particularly if they were undocumented immigrants).

## Effects of coping with COVID-19 on mental well-being

Given the disproportionate effects of COVID-19 on the Black population, it would be logical to expect Blacks to exhibit the greatest losses in mental health and other dimensions of well-being. Yet, that is not what we found; Black respondents–and particularly low income ones—maintained higher levels of resilience–more optimism and better mental health–than White respondents *throughout the course of the survey*. While this result is in line with those of pre-COVID-19 studies that find high levels of optimism and resilience among Blacks compared to those of other races, is it remarkable that it held throughout the pandemic [26].

We would normally anticipate a negative association between COVID-19 related concerns and well-being during the pandemic–that is, the higher COVID-19 related concerns, the lower life satisfaction/optimism, and the lower mental health. Yet they are remarkably consistent with patterns that we have previously found in the well-being of different race and income cohorts in the face of deaths of despair. Using over 1 million responses over five years in Gallup data for the U.S., one of us (Graham, with Sergio Pinto) found large gaps in optimism and reported stress across poor Black and White individuals, with the former almost three times as likely to be a point higher on the 11-point optimism scale and 50 percent less likely to report experiencing stress the previous day than poor Whites (poor Hispanics again fall in between the two groups on the same markers). This finding is a lasting one, holding from 2010 until the present and now throughout the course of the pandemic [26].

While Black respondents are more optimistic than Whites in both sets of data, the largest differences are between poor Blacks and poor Whites. These patterns are reflected in those in C.D.C. data on deaths of despair, in which Blacks and Hispanics are much less likely to die from these deaths than are Whites.

The reasons for this resilience are complex. They include a historical trajectory of overcoming adversity, strong community ties, and continued belief in the promise of education at a time that it has faded among low-income Whites. As a result, Blacks and Hispanics are gradually narrowing gaps in education and in life expectancy with Whites. Poor Whites, meanwhile, have fallen behind in absolute terms compared to wealthy Whites and in relative terms compared to minorities; losses that are reflected in their high levels of despair. Historically, meanwhile, optimism among Blacks began to increase in the 1970s, when civil rights improved, and began to fall among less than college-educated White men around the same time (coinciding with the first declines in manufacturing) [39]. It seems that the same traits that drive minority resilience may also be protective of well-being and mental health in the context of the pandemic.

Deaths of despair are, at least at some level, preventable because they are desperation-related behaviors. In contrast, COVID-19 is an exogenous shock–largely unrelated to individual behavior–that has disproportionately affected the Black population. Despite such disparities, in both of our studies the Black respondents have displayed high levels of hope and resilience. It appears that the same traits that drive minority resilience might also be protective of well-being and mental health in the context of the pandemic. While that is the hope, we still do not know what the long-run effects of the COVID-19 pandemic will be. Excess deaths in 2020 (both due to COVID-19 and to many other conditions, including pre-existing conditions and increases in drug overdoses) were highest among Blacks, and there are many reports of increased anxiety, depression, and related outcomes, particularly among the young, with minority teens and young adults in particular experiencing increases from past (relatively lower) levels [40, 41].

These deep differences in resilience have implications for the long-term mental health effects of different population groups in the face of the unprecedented challenge that COVID-

19 presents to the health and well-being of our society. Better understanding these differences–and the lessons that stem from those population cohorts with the most resilience–can, in the end, lead to lessons that may help bolster the mental health and coping skills and behavioral responses of vulnerable groups during uncertain times.

## Supporting information

**S1 Appendix. Survey items used in the analysis.**
(DOCX)

## Author Contributions

**Conceptualization:** Carol Graham, Yung Chun, Bartram Hamilton, Stephen Roll, Wilbur Ross, Michal Grinstein-Weiss.

**Data curation:** Yung Chun, Stephen Roll.

**Formal analysis:** Yung Chun.

**Funding acquisition:** Michal Grinstein-Weiss.

**Methodology:** Carol Graham, Yung Chun.

**Writing – original draft:** Carol Graham.

**Writing – review & editing:** Yung Chun, Bartram Hamilton, Stephen Roll, Wilbur Ross.

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
