## [Decision Letter · Decision Letter 0]

11 Nov 2021

PONE-D-21-27864Coping with COVID-19: Differences in Hope, Resilience, and Mental Well-being across U.S. Racial GroupsPLOS ONE

Dear Dr. Graham,

Thank you for submitting your manuscript to PLOS ONE. After careful consideration, we feel that it has merit but does not fully meet PLOS ONE’s publication criteria as it currently stands. Therefore, we invite you to submit a revised version of the manuscript that addresses the points raised during the review process.

We look forward to receiving your revised manuscript.

Kind regards,

Sanjay Kumar Singh Patel, Ph.D.

Academic Editor

PLOS ONE

Journal Requirements:

2. Please provide additional details regarding participant consent. In the Methods section, please ensure that you have specified (1) whether consent was informed and (2) what type you obtained (for instance, written or verbal). If your study included minors, state whether you obtained consent from parents or guardians. If the need for consent was waived by the ethics committee, please include this information.

There was not a specific funding source for this article. The survey was funded by the School of Social Policy at Washington University in St. Louis.

The authors have no competing interests. 

Reviewers' comments:

Reviewer's Responses to Questions

**Comments to the Author**

1. Is the manuscript technically sound, and do the data support the conclusions?

Reviewer #1: Yes

Reviewer #2: Yes

2. Has the statistical analysis been performed appropriately and rigorously? 

Reviewer #1: Yes

Reviewer #2: Yes

3. Have the authors made all data underlying the findings in their manuscript fully available?

Reviewer #1: Yes

Reviewer #2: No

4. Is the manuscript presented in an intelligible fashion and written in standard English?

Reviewer #1: Yes

Reviewer #2: Yes

5. Review Comments to the Author

Reviewer #1: In this paper entitled "Coping with COVID-19: Difference in hope, resilience and mental well-being across U.S. racial groups", the author examines how the COVID-19 pandemic impacts health and mental well-being across different racial groups. The studies have used advanced statistical techniques like logistic regression and fixed effect modeling to estimate COVID-19 impact on health and mental well-being. The study found that Black and Hispanics remain more resilient and optimistic than their white counterparts despite extreme income and health disparities before and during the COVID-19 outbreak. The most significant difference in resilience, optimism, and better mental health is between poor black and white poor. These results help to understand the factors involved to preserve mental health in the face of public health and other large cities crises. The present manuscript is well written and easy to understand. This manuscript is an excellent study that draws attention to adverse mental health and well-being during the COVID-19 pandemic. It also allowed understanding how to reduce the social and economic consequences of adverse mental health, which improves health and well-being during the pandemic. There are no technical flaws in the manuscript to reject it. Therefore, it should be considered for the publication.

Reviewer #2: This is an excellent paper by Graham and colleagues that could not be more timely and relevant to the current discourse surrounding health inequities in the COVID-19 era. The methodology is sound and the results are well presented. Additionally, the principal findings of increased resilience among and optimism among Blacks and Hispanics could certainly shape the provision of mental health and support services for these groups going forward. I would recommend this paper for publication following minor revisions. I have very few comments (please see below).

General

• Authors should maintain consistency in the use of “COVID-19” or “COVID” throughout the paper

Abstract

• Line 30 – the word “if” is missing from after the word “explore”

Introduction

• Line 71 – these comorbidities are risk factors for poorer COVID-19 outcomes (not for COVID-19 acquisition)

Methods

• Authors could consider including the survey as an appendix/supplementary material to complement the description of the tool in the methods section

Results

• Figure 6 appears to be the same graph twice (both low-income results). The mid/high-income graph is missing

Discussion

• Did the authors identify any limitations of their study?

---

## [Author Response · Author response to Decision Letter 0]

1 Feb 2022

On behalf of myself and co-authors, we thank you for the attention to our manuscript entitled “Coping with COVID-19: Differences in Hope, Resilience, and Mental Well-being across U.S. Racial Groups”, the positive and helpful comments of both reviewers, and the opportunity to re-submit the paper. 

First, we have adapted the formatting to meet the PLOS ONE standards. Second, in terms of data, we have created a GitHub access link, which provides the raw data files and variables that we used for our analysis in this paper, as well as the codes (https://github.com/SocialPolicyInstitute/SEICS/tree/main/PLoS-ONE). 

Third, although this paper did not receive any direct funding from any source, the collection of the data used in this paper was funded by grants to the SPI at Wash U from the Mastercard Foundation, the JPMorgan Chase Foundation, the Annie E. Casey Foundation, and Centene Corporation. These funders had no role in study design, data collection and analysis, decision to publish, or preparation of the manuscript. Fourth, the authors have declared that no competing interests exist. 

Fifth, we have adapted the methods statement to include that the survey was submitted to the Washington University at St. Louis IRB (ID: 202004100) and was approved. This was not a requirement as it is an on-line internet survey with voluntary participation, but we sought the additional approval. We received written consent. Even though it was exempt, we still got written (online) consent from all research participants before they entered the survey. The survey was also limited to adults.

Fifth, as required we have checked the reference section to ensure that all references in the list are referenced in the text. We have also uploaded the figures to the PACE tool as recommended. 

In terms of responses to Reviewer 2’s more detailed comments, we have: standardized our reference to COVID-19 throughout the manuscript; fixed the missing panel in Figure 6; fixed the typo on line 30; and made the point on line 71 that the co-morbidities we refer to are risk factors for poor COVID-19 outcomes, not for COVID-19 acquisition. Finally, we have included the survey questionnaire as supplementary material and reference that in the methods section.

Again, we much appreciate the chance to submit a revised manuscript and hope that we have fully addressed the questions above. We are pleased to have the opportunity to publish in PLOS ONE.

---

## [Decision Letter · Decision Letter 1]

12 Apr 2022

Coping with COVID-19: Differences in Hope, Resilience, and Mental Well-being across U.S. Racial Groups

PONE-D-21-27864R1

Dear Dr. Graham,

We’re pleased to inform you that your manuscript has been judged scientifically suitable for publication and will be formally accepted for publication once it meets all outstanding technical requirements.

Kind regards,

Giuseppe Carrà, PhD

Academic Editor

PLOS ONE

Additional Editor Comments (optional):

Reviewers' comments:

Reviewer's Responses to Questions

**Comments to the Author**

1. If the authors have adequately addressed your comments raised in a previous round of review and you feel that this manuscript is now acceptable for publication, you may indicate that here to bypass the “Comments to the Author” section, enter your conflict of interest statement in the “Confidential to Editor” section, and submit your "Accept" recommendation.

Reviewer #1: All comments have been addressed

2. Is the manuscript technically sound, and do the data support the conclusions?

Reviewer #1: Yes

3. Has the statistical analysis been performed appropriately and rigorously? 

Reviewer #1: Yes

4. Have the authors made all data underlying the findings in their manuscript fully available?

Reviewer #1: Yes

5. Is the manuscript presented in an intelligible fashion and written in standard English?

Reviewer #1: Yes

6. Review Comments to the Author

Reviewer #1: In this paper entitled "Coping with COVID-19: Differences in Hope, Resilience, and Mental Well-being across U.S. Racial Groups", the authors have addressed all the comments and have no technical deficiency for rejection. The paper is eligible for acceptance in the journal.

7. PLOS authors have the option to publish the peer review history of their article (what does this mean?). If published, this will include your full peer review and any attached files.

Reviewer #1: **Yes: **Aditya Kumar Sharma

---

## [Editor Report · Acceptance letter]

29 Apr 2022

PONE-D-21-27864R1 

Coping with COVID-19: Differences in hope, resilience, and mental well-being across U.S. racial groups 

Dear Dr. Graham:

I'm pleased to inform you that your manuscript has been deemed suitable for publication in PLOS ONE. Congratulations! Your manuscript is now with our production department. 

Kind regards, 

on behalf of

Dr. Giuseppe Carrà 

Academic Editor

PLOS ONE